# A Rat Study to Evaluate the Protein Quality of Three Green Microalgal Species and the Impact of Mechanical Cell Wall Disruption

**DOI:** 10.3390/foods9111531

**Published:** 2020-10-24

**Authors:** Yanwen Wang, Sean M. Tibbetts, Fabrice Berrue, Patrick J. McGinn, Scott P. MacQuarrie, Anil Puttaswamy, Shane Patelakis, Dominique Schmidt, Ronald Melanson, Sabrena E. MacKenzie

**Affiliations:** 1Aquatic and Crop Resource Development Research Center, National Research Council of Canada, 550 University Avenue, Charlottetown, PE C1A 4P3, Canada; Anil.Puttaswamy@nrc-cnrc.gc.ca (A.P.); Dominique.Schmidt@nrc-cnrc.gc.ca (D.S.); 2Aquatic and Crop Resource Development Research Center, National Research Council of Canada, 270 Sandy Cove Road, Ketch Harbor, NS B3V 1K9, Canada; Sean.Tibbetts@nrc-cnrc.gc.ca (S.M.T.); patrick.mcginn@nrc-cnrc.gc.ca (P.J.M.); scott.macquarrie@nrc-cnrc.gc.ca (S.P.M.); Shane.Patelakis@nrc-cnrc.gc.ca (S.P.); Ronald.Melanson@nrc-cnrc.gc.ca (R.M.); 3Aquatic and Crop Resource Development Research Center, National Research Council of Canada, 1411 Oxford Street, Halifax, NS B3H 3Y8, Canada; Fabrice.Berrue@nrc-cnrc.gc.ca (F.B.); Sabrena.MacKenzie@nrc-cnrc.gc.ca (S.E.M.)

**Keywords:** amino acid score, microalgae, protein digestibility, protein digestibility-corrected amino acid score, rat

## Abstract

The present study was conducted to evaluate the protein quality of microalgae species *Chlorella vulgaris* (CV), *Chlorella sorokiniana* (CS), and *Acutodesmus obliquus* (AO) and assess the impact of mechanical cell wall disruption. Male Sprague–Dawley rats, around 156 g after adaptation, were placed in metabolic cages and fed experimental diets that were either protein-free or contained 10% protein solely from one of the undisrupted or disrupted CV, CS, and AO. After 3 days, feces were collected for a period of 5 days and analyzed together with diet samples for crude protein contents. Apparent protein digestibility, true protein digestibility, amino acid score, and protein digestibility-corrected amino acid score were calculated. In vitro protein digestibility was measured using the pepsin–pancreatin method and the in vitro protein digestibility-corrected amino acid score was calculated. The crude protein contents of CV, CS, and AO were 53.5, 50.2, and 40.3%, respectively. The amino acid score of the first limiting amino acid was 1.10, 1.27, and 0.86, true protein digestibility was 64.7, 59.3, and 37.9% and protein digestibility-corrected amino acid score was 0.63, 0.64, and 0.29, respectively, for CV, CS, and AO. Mechanical cell disruption significantly improved protein digestibility without a substantial impact on the amino acid profile and score, resulting in the increase of protein digestibility-corrected amino acid score to 0.77, 0.81, and 0.46, respectively, for disrupted CV, CS, and AO. There was a strong correlation between in vitro protein digestibility and apparent protein digestibility (*r* = 0.986), and also between in vitro protein digestibility-corrected amino acid score and in vivo protein digestibility-corrected amino acid score (*r* = 0.994). The results suggest that the CV and CS are acceptable sources of protein for humans and animals and quality can be markedly improved by mechanical cell wall disruption. Additionally, in vitro protein digestibility measured using the pepsin–pancreatin method may be used to screen protein product candidates, save animals, reduce cost, and accelerate product development.

## 1. Introduction

A growing global population, combined with other factors such as changing socio-demographics, has placed increased pressure on the world’s resources to provide not only more but also different types of food [1]. In order to sustain the current and future world population, food production needs to be increased intensively, which raises concerns about sustainability and food security. In addition, factors such as rising income, increasing urbanization, aging population, and increasing awareness of the health impacts of different nutrients are continuously shifting consumption patterns, where increased protein consumption is of particular interest [2].

Foods of animal origin have no doubt played a critical role in human food security and provided high-quality proteins and a variety of micronutrients such as vitamin and minerals, which can be difficult to obtain in adequate quantities from plant-based foods alone [3,4,5]. However, several factors determine the overall impact of particular sources of food on food security and sustainability [6]. It is understood that animal feed rations contain ingredients that can serve as human food components and that land used for producing animal feed is also suitable for human food production. Furthermore, animals have a low efficiency of converting feed into human food [3]. Therefore, the intensification of food production requires a balance with the environmental impact and efficiency, with livestock production meriting particular scrutiny [7]. One of a host of emerging strategies worldwide is to develop novel sources of high-quality plant-based proteins. In this regard, microalgae have certain advantages over traditional crops. They are considered renewable and provide a long-term sustainability of food production because of their favorable carbon, water, and arable land footprints compared to terrestrial crops [8,9]. There are numerous microalgae species known to science, while the majority have not been explored for their chemical compositions and nutritional values [10]. The limited information available indicates that microalgae species differ largely in the composition of nutrients and bioactive metabolites, offering great potential and flexibility to be used as a platform for producing interesting compounds and functional ingredients, including proteins [9,11,12].

Past research has examined the nutrient composition of selected microalgae species [13,14,15], with some of them being evaluated in vitro for their uses in ruminant livestock and monogastric animals [16,17,18]. *Chlorella vulgaris* was studied for protein and amino acid digestibility in farmed Atlantic salmon [19]. However, the applicability of microalgae protein to human food is largely unknown, because specific methods and markers required to assess protein quality for human nutrition have not been widely applied to microalgae. Protein content and amino acid composition differ widely across microalgal species [20]. It is also well understood that the cell wall of many microalgae species is not readily broken up by the physical, chemical and biochemical actions of a typical monogastric digestive system, which thus affects protein release and digestibility. Indeed, in vitro studies showed that the protein digestibility of microalgae was low compared to animal sources of protein and different between microalgae species [21,22]. Another study conducted recently showed that in vitro protein digestibility varied, to a large extent, in 16 microalgae products of four genera, including *Arthrospira, Chlorella, Nannochloropsis,* and *Phaeodactylum* and was improved by sonication cell rupture [20]. The present study was conducted in a rat assay to determine the protein quality for human nutrition of three different microalgal species selected for high protein content and the potential improvement by mechanical cell rupture. 

## 2. Materials and Methods 

### 2.1. Biomass Production of Three Green Microalgal Species

Three freshwater Chlorophytic microalgal species were chosen for high-protein contents from the microalgae library of the National Research Council of Canada (NRC) in Ketch Harbor, NS, Canada. They were *Acutodesmus obliquus*/NRC isolate SMC- 6F (AO), *Chlorella vulgaris*/NRC isolate AB12-A-U-BBM (CV), and *Chlorella sorokiniana*/NRC isolate SMC-15M (CS), respectively. Approximately 1.2 kg freeze-dried biomass of each species were produced at the NRC’s Algal Research and Feed facility (Ketch Harbor, NS, Canada). Briefly, microalgae were grown in 1000 L Brite Box^TM^ photobioreactors (described in [13]) in batch mode and harvested every week for 3 weeks (wk) in the late exponential growth phase after an initial 10 days of growth. The photobioreactors were maintained at 25 °C and pH 7.2 through the addition of CO_2_ via a pH-stat. Sterilized *f/2* medium was fed in batch every 2–3 days to ensure that nutrient levels remained replete. Algal biomass was separated from bulk media using a continuous flow centrifuge (model Z101; CEPA Carl Padberg Zentrifugenbau GmbH., Lahr, Germany) at 25 °C, with a flow rate of 1000 L hr^−1^ and a speed of 14,000 rpm (5500× *g*), and subsequently lyophilized using a pilot-scale freeze dryer (model 35 EL; VirTis Company, Gardiner, NY, USA). Then, the dried biomass samples were stored at −80 °C until further processing. 

### 2.2. Mechanical Cell Rupture of Microalgae Biomass

The biomass samples were processed following the procedures established previously [19]. Briefly, the freeze-dried algal biomasses of CV, CS, and AO were pulverized (to pass through a 500 µm screen) at 10,000 rpm using a laboratory ultra-centrifugal mill (model ZM200; Retsch GmbH., Haan, Germany) equipped with a Retsch pneumatic auto-feeder (model DR100). Half of each algal biomass was stored at −80 °C, and the other half was reconstituted with distilled water at an initial ratio of 1:6 (algae:water, *w*/*v*). The AO was further diluted to 1:7 (*w*/*v*), as it was too viscous for effective flow through the homogenization process. Reconstituted algal biomass was cell-ruptured by high-pressure homogenization using a bench-top Microfluidizer^®^ (model M-110P; Microfluidics International Corporation, Westwood, MA, USA). The operating parameters were 25,000 psi (1724 bar) at flow rates ranging between 25 and 75 mL min^−1^ (depending on species and pass number) through two in-series ‘Z’ configuration interaction chambers including a 200 μm module (model H30Z diamond) and an 87 μm module (model G10Z diamond). The material was passed through the instrument twice. The retention time through the instrument is approximately 5–10 s without temperature control on the instrument. To minimize possible thermal damage related to high shear forces, the product was passed through a chilling coil surrounded by crushed ice immediately upon exit of the 87 μm module. Additionally, microfluidized product was collected directly into a chilled container. Our previous work demonstrated that a double pass under the same processing conditions is sufficient for near-complete cell wall disruption of *Chlorella* microalgae, although it cannot be strictly controlled [19]. The final microfluidized product (after double pass) was lyophilized for 96 h at a low shelf temperature (<5 °C) to a final moisture content of <5% and reground using a coffee grinder for 20–30 s, giving a particle size of ≤ 500 µm. The mechanically ruptured CV, CS, and AO were referred to as CVMR, CSMR, and AOMR, respectively, and stored at −80 °C.

### 2.3. Proximate Analysis of Microalgal Biomass and Casein Samples

The CV, CS, AO, CVMR, CSMR, and AOMR were each analyzed in triplicate for their proximate composition using published methods [13]. Specifically, moisture and ash contents were determined gravimetrically by drying in an oven at 105 °C for 18 h and by incineration in a muffle furnace at 550 °C for 18 h, respectively. Nitrogen (N) contents were determined using a LECO FP-528 nitrogen analyzer and EDTA with certified 9.56% nitrogen as the internal standard (LECO Corporation, St. Joseph, MI, USA). The furnace temperature was set at 950 °C, with ultra-pure oxygen as the combustion gas and ultra-pure helium as the carrier gas. The content of crude protein was calculated as elemental N × 6.25. Crude lipids were extracted using an HPLC-grade chloroform and methanol mix (2:1 *v*/*v*) at 150 °C for 82 min on a Soxtec automated extraction system (model 2050; FOSS North America, Eden Prairie, MN, USA) [23]. Carbohydrate contents were determined by the colorimetric method described by Dubois et al. [24] using phenol and sulfuric acid following acid hydrolysis (2.5 M HCl at 95 °C for 3 h) according to the method reported by Sukenik et al. [25]. Final results were determined against a dextrose standard curve (0–100 μg mL^−1^, D-glucose, Cat. G5400, solid, >99% pure; Sigma-Aldrich, Oakville, ON, Canada). Starch contents were determined by the α-amylase and amyloglucosidase method [26] using a total starch assay kit (K-TSTA; Megazyme International Ireland Ltd., Wicklow, Ireland) accepted by the Association of Official Agricultural Chemists (Method 996.11) and American Society of Cereal Chemistry (Method 76.13). Crude fiber contents were estimated using the ANKOM filter bag technique according to the American Oil Chemists’ Society (Method Ba 6a- 05). Casein (Dyets Inc., Bethlehem, PA, USA) was used as a standard protein in the animal study, and the protein content was analyzed using the same method as for the analysis of microalgae samples.

### 2.4. Amino Acid Analysis of Microalgae Samples and Casein

Amino acid contents were analyzed using the Waters Pico-Tag reversed phase-HPLC method [27] through a contract service (VALORĒS Research Institute, Shippagan, NB, Canada). Samples were hydrolyzed with 6N HCl on the Waters Pico-Tag workstation except for tryptophan and cysteine, which were prepared using different methods prior to HPLC analysis. For tryptophan analysis, the samples were hydrolyzed with 4 M methanesulfonic acid containing 0.2% (*w*/*v*) tryptamine hydrochloride (Sigma-Aldrich, Oakville, ON, Canada). Performic acid oxidation was used for the determination of cysteine and cysteine residues [28]. The samples were centrifuged, and the supernatants were dried and derivatized in MeOH:H_2_O:TEA:PITC (7:1:1:1 *v*/*v*/*v*/*v*). The supernatants were collected following centrifugation and reconstituted and transferred into the autosampler inserts. The Waters Acquity UPLC BEH C18 column (2.1 × 100 mm) equipped with a Waters Acquity TUV detector was used. The mobile phase was consisted of buffer A (0.14 M sodium acetate, 0.05% triethylamine, pH 6.05, 6% acetonitrile) and buffer B (60% acetonitrile) with 6 min gradient. The column temperature was set at 48 °C and signals were detected at 254 nm. Data were collected, stored, and processed using Waters Empower 3 Chromatography software (Waters Ltd., Mississauga, ON, Canada). The external standard was a Pierce Amino Acid Standard H (ThermoFisher Scientific, Ottawa, ON, Canada) with the addition of 50 pM each of norleucine, hydroxyproline, taurine, alpha-aminobutyric acid, and ornithine (Sigma-Aldrich, Oakville, ON, Canada). The internal standard was 200 pM of norleucine. The content of amino acids is presented as mg g^−1^ protein (the amount of total amino acids). Amino acid score of the individual essential amino acids was calculated using the following equation:Amino acid score = mg of amino acid in 1 g of test protein/mg of the same amino acid in 1 g of reference protein(1)

The reference protein was the recommended pattern of amino acids for 3–10 years old children by the FAO/WHO/UNU, 2007 [29]. 

### 2.5. Animal Study

The animal study was carried out in 5 batches, with 16 rats per batch and 2 rats per group in each batch. The assay protocol was based on the Association of Official Agricultural Chemists Official Method 991.29 [30]. Briefly, male Sprague–Dawley rats, approximately 70 g, were purchased from Charles River (Saint-Constant, QC, Canada) and housed individually in rat cages after arrival. They were allowed to become acclimated to the new environment for 8 days except for the first batch, which had 14 days of acclimatize, as a pilot test was performed for the palatability of test products prior to the formal experiment. During the acclimation, rats were provided with free access to water and regular laboratory chow (Laboratory rodent diet 5001; LabDiet, St. Louis, MO, USA). Then, rats were randomized based on the body weight into groups and transferred into metabolic cages individually and fed a rodent AIN-93G diet modified to contain zero protein (protein-free diet) or 10% protein in which one of CV, CS, AO, CVMR, CSMR, and AOMR was the sole source of protein. A diet containing 10% casein protein as the sole source of dietary protein was used as a standard protein control. The protein-free diet was used to estimate the endogenous metabolic protein loss, which is essential for the calculation of true protein digestibility (TPD). The 10% protein diets were adjusted for fat, starch, fiber, and ash based on the results of proximate analysis by deducting the amount of each provided by the test products from lard, corn starch, cellulose, and mineral mix, respectively, in the respective diets so that all diets had similar contents of macronutrients. The diet composition is provided in the Appendix A. After 3 days on the test diets, feces were collected from individual rats for a period of 5 days, with food intake being recorded daily. The feces of each rat were pooled and kept at −80 °C until further processing (Scheme 1). The animal use and experimental procedures were approved by the Joint Animal Care and Research Ethics Committee of the National Research Council Canada and the University of Prince Edward Island in Charlottetown (ACC No. 19–053 and File No. 6008325). The study was conducted in accordance with the guidelines of the Canadian Council on Animal Care.

### 2.6. Protein Analysis of Diet and Fecal Samples

Fecal samples were freeze-dried, weighed, and analyzed in duplicate, together with diet samples, for protein contents using the same method described above in the proximate analysis. 

### 2.7. Protein Digestibility and Protein Digestibility-Corrected Amino Acid Score 

Apparent protein digestibility (APD), TPD, and protein digestibility corrected amino acid score (PDCAAS) were calculated, respectively, using the following equations: APD% = (dietary protein intake − fecal protein loss)/dietary protein intake × 100(2)
TPD% = (dietary protein intake − (fecal protein loss − endogenous protein loss))/dietary protein intake × 100(3)
PDCAAS = True protein digestibility × amino acid score of the first limiting amino acid(4)

### 2.8. In Vitro Protein Digestibility and In Vitro Protein Digestibility-Corrected Amino Acid Score

In vitro protein digestibility (IVPD) of microalgae samples was determined in duplicate using a pepsin–pancreatin method developed for farmed salmonids [13]. Briefly, two-phase gastric and intestinal digestions were performed by weighing 250 mg of test sample in a 50 mL tube containing 12.5 mL PBS, 5 mL of 0.2 M HCI, and 250 µL of chloramphenicol solution. After warming up to 39 °C, 500 µL of 25 mg mL^-1^ pepsin in 0.2 N HCI, pH 1.0 were added. After incubation for 4.5 h at 39 °C with head-over-heel agitation on a tube rotator, 5 mL PBS and 2.5 mL of 0.6 N NaOH were added to stop the reaction. After the addition of 500 µL of 100 mg mL^−1^ pancreatin in 0.05 M tris and 0.0115 M CaCI_2_ buffer (pH = 8), the tubes were incubated for 18 h at 39 °C while shaking. The reaction was stopped by adding 2.4 mL of 20% sulfosalysilic acid. The solution was filtered with a Waterman filter paper (G/F/A, 1.6 µm; Sigma-Aldrich), and the residue was analyzed for nitrogen content. In vitro protein digestibility was calculated as:IVPD% = (protein in initial sample − protein in digested residue)/protein in initial sample × 100.(5)

In vitro protein digestibility-corrected amino acid score (IVPDCAAS) was calculated as the product of the first limiting amino acid score and in vitro protein digestibility [31,32].

### 2.9. Data Analysis

In vivo apparent protein digestibility and true protein digestibility were analyzed, respectively, for differences among microalgae species and the effects of mechanical cell rupture with a two-way ANOVA using SAS software ver. 9.4 (SAS Institute, Cary, NC, USA). When a significant effect was detected, the difference among the species and the effect of mechanical cell rupture were further determined using the least squares means test adjusted to Tukey. The results are presented as means ± S.E.M. (*n* = 10), with the significance level being set at *p* < 0.05. Other data are presented as the average of replicates as specified in the footnote of tables. 

## 3. Results

### 3.1. Nutrient Composition of Three Microalgal Species 

Nutrient composition differed largely among the three microalgal species (Table 1). The CV had the highest protein, fiber, and ash contents but the lowest lipids, carbohydrate, and starch. In contrast, the AO had the lowest protein, ash, and fiber contents but the highest content of carbohydrate. The CS had the highest starch content and similar content of lipids with the AO, with the content of other nutrients intermediates between the CV and AO. 

### 3.2. Effect of Mechanical Cell Rupture on the Nutrient Content of Three Microalgae Species

Mechanical cell rupture had no effect on protein, carbohydrate, or ash content, while a substantial effect was found on lipid, starch, and fiber contents (Table 1). The CVMR had 58.5% higher lipid content than the CV. As a result, the CVMR became the one that had the highest lipid content, while the CV had the lowest lipid content among the disrupted and undisrupted three microalgae species. A slight increase of lipid content was noted in the CSMR and AOMR as compared with their counterparts, CS and AO. Starch content was increased by 130.5% in the AOMR compared to the AO. An increase of 15.6% was observed in the CSMR relative to the CS, while similar contents were noted between the CVMR and CV. The fiber content was increased by 1330% in the AOMR relative to the AO and by 160% in the CSMR compared to the CS, while it slightly decreased in the CVMR as compared with the CV. 

### 3.3. Amino Acid Composition of Three Microalgal Species

The amino acid content of three microalgal species is presented in Table 2. Regarding the essential amino acids, among the three microalgae species, the CS had the highest and the AO had the lowest contents of histidine, ranging from 14.1 mg g^−1^ protein to 20.5 mg g^−1^ protein. The content of threonine ranged from 42.9 mg g^−1^ protein in the CV to 58.6 mg g^−1^ protein in the AO. The CV and CS had close contents of cysteine, with the highest content being found in the AO. The CS had the highest lysine content, followed by the CV and then AO. The CV had the highest content of tryptophan, while the CS and AO had close amounts. The contents of methionine, isoleucine, and phenylalanine were the lowest in the CS but similar between the CV and AO. The three species had similar contents of tyrosine and valine. In comparison to casein, histidine and isoleucine were lower while tryptophan and threonine were higher in the three microalgae species. For more details and the amino acid contents in the biomass of the disrupted and undisrupted three microalgae species, see Appendix A.

### 3.4. Impact of Mechanical Cell Rupture on the Amino Acid Content of Three Microalgal Species

Mechanical cell rupture did not cause significant changes to the amino acid profile compared to undisrupted biomass (Table 2). For each species, the amino acid contents of disrupted biomass were highly correlated with that of undisrupted biomass, with *r* = 0.997 (*p* < 0.0001) for CV vs. CVMR, *r* = 0.998 (*p* < 0.0001) for CS vs. CSMR, and *r* = 0.994 (*p* < 0.0001) for AO vs. AOMR, respectively. Significant correlations (*p* < 0.0001) were observed between casein and each of the three disrupted microalgae species samples. 

### 3.5. Amino Acid Score of Three Microalgae Species and the Impact of Mechanical Cell Rupture

Amino acid scores of undisrupted and disrupted three microalgae species are presented in Table 3. The score of histidine was the lowest in the AO and AOMR, while the score of isoleucine was the lowest in the CS and CSMR. The score of histidine was the lowest in the CV, but the score of methionine plus cysteine was the lowest score in the CVMR. However, the scores of histidine and methionine plus cysteine were almost the same in the CVMR. Mechanical cell rupture did not show a significant effect on the amino acid composition and thus did not change considerably the amino acid score as compared to the undisrupted biomass samples. There were significant correlations (*p* < 0.0001) of the essential amino acid contents between the CV and CVMR (*r* = 0.996), between the CS and CSMR (*r* = 0.998), and between the AO and AOMR (*r* = 0.990), respectively. The abundance of the first limiting amino acid determines the utilization efficiency of all essential amino acids for protein synthesis; thus, the amino acid score of the first limiting amino acid is a key factor in assessing protein quality. It was 1.10 for the CV and CVMR, although the first limiting amino acid was histidine for the CV and changed to methionine plus cysteine for the CVMR. Isoleucine was the first limiting amino acid of the CS and CSMR and had the amino acid score of 1.16. Mechanical cell rupture did not change the amino acid score of the first limiting amino acid in these two microalgae species. Histidine was the first limiting amino acid for the AO and AOMR and the amino acid score was 0.86 and 0.76, respectively, which were markedly lower than the CV, CVMR, CS, and CSMR. The decrease of histidine score of the AOMR might be a result of analytical error, as the score of other amino acids was not decreased. In casein, the first limiting amino acid was methionine plus cysteine with the amino acid score of 1.26, which was slightly higher than that of the CV, CVMR, CS, and CSMR, but much higher than that of the AO and AOMR. The secondary limiting amino acid was leucine for casein, isoleucine for CV, methionine plus cysteine for CS, histidine for CVMR and CSMR, isoleucine for AO, and lysine for AOMR, respectively.

### 3.6. In Vivo and In Vitro Protein Digestibility of Three Microalgal Species

The apparent protein digestibility, true protein digestibility, and in vitro protein digestibility of undisrupted and disrupted three microalgae species are presented in Table 4. The standard protein casein had an apparent protein digestibility of 94.4% and true protein digestibility of 98.4%, which was within the range of values reported in several previous studies [33,34,35,36,37,38,39], demonstrating that the study protocol and analytical methods for measuring protein digestibility were reliable. The same methods were used to determine the protein digestibility of microalgae samples. The CV had the highest values of protein digestibility, while the AO had the lowest. The apparent protein digestibility was 9.6% higher (*p* < 0.05) in the CV than in the CS and 85.9% higher (*p* < 0.05) than in the AO. The CS had 68.1% higher (*p* < 0.05) apparent protein digestibility than the AO. Similarly, the CV showed 9.1% higher (*p* < 0.05) true protein digestibility than the CS and 71.0% higher (*p* < 0.05) than the AO, while the true protein digestibility of the CS was 56.6% higher (*p* < 0.05) than that of the AO. The protein digestibility of the AO was much lower compared to the CV and CS, while the difference between the CS and CV was small. The in vitro protein digestibility of three microalgae species was close to the values of apparent protein digestibility or true protein digestibility. 

### 3.7. Impact of Mechanical Cell Rupture on the in Vivo and in Vitro Protein Digestibility of Microalgal Species

As shown in Table 4, mechanical cell rupture showed a significant improvement on both the apparent protein digestibility and true protein digestibility of three microalgae species. A large increase was observed in the AOMR compared to the AO, while moderate increases were noted for the CVMR and CSMR relative to their undisrupted counterparts, CV and CS. As a result, the difference between the CVMR, CSMR, and AOMR became much smaller as compared with differences between the CV, CS, and AO. Either apparent protein digestibility or true protein digestibility did not differ between the CVMR and CSMR, while the AOMR had lower (*p* < 0.05) apparent protein digestibility and true protein digestibility than the CSMR or CSMR. Similar improvements were observed in the in vitro protein digestibility of the three microalgae samples. A further analysis of six samples in a pool revealed significant correlations between in vitro protein digestibility and apparent protein digestibility (*r* = 0.986, *p* = 0.0003) (Figure 1A) or true protein digestibility (*r* = 0.985, *p* = 0.0003) (Figure 1B).

### 3.8. PDCAAS of Three Microalgal Species and the Impact of Mechanical Cell Rupture

The protein digestibility-corrected amino acid score of casein was 1.24 and truncated to 1.00 according the method defined by the AFO/WHO (Table 4). The CV and CS had similar values of protein digestibility-corrected amino acid score, 0.63 and 0.64, while the AO had a protein digestibility-corrected amino acid score of 0.29, which was 54.1% and 54.8% lower than the CV and CS, respectively. Mechanical cell rupture improved the protein digestibility-corrected amino acid score, resulting in 16.7, 26.6, and 58.6% of increase in the CVMR, CSMR, and AOMR compared to their counterparts, CV, CS, and AO, respectively. Again, it was similar between the CVMR and CSMR, whereas the value of the AOMR was 40.8 and 43.6% lower than that of the CVMR and CSMR, respectively. A lower protein digestibility-corrected amino acid score of the AO was a result of lower values of both true protein digestibility and first limiting amino acid score. The weight gain of each group corresponded with the protein quality, where rats on the casein diet had much higher weight gain than rats given either of the three species of microalgae biomass as the sole protein source. Among the three microalgal species, rats given the CS and CSMR had the highest weight gain compared with rats provided with CV, CVMR, AO, and AOMR, respectively. Rats on the AO diet had the lowest and negative weight gain and dramatically improved in those given the mechanically ruptured AO. For each biomass sample of the three microalgae species, undisrupted and disrupted, the in vitro protein digestibility-corrected amino acid score was close to the in vivo protein digestibility-corrected amino acid score, and a strong correlation was found between both parameters (*r* = 0.994, *p* < 0.0001; Figure 2).

## 4. Discussions

There are many different sources of protein products, which differ not only in the protein quantity but also the quality [31,40,41]. Protein nutrition has been assessed worldwide based on the total protein intake in combination with protein quality, and the latter is the bottleneck for developing new protein products and alternatives such as plant-based proteins [31,32,36]. Protein quality is determined by protein digestibility or amino acid digestibility combined with the content of each essential amino acid, in particular the first limiting amino acid, relative to the recommendation, e.g., the reference protein amino acid pattern [29]. The present study showed that protein content, protein digestibility, amino acid score, and protein digestibility-corrected amino acid score were close between microalgae CV and CS, whereas microalgae AO had much lower values than the CV and CS. Mechanical cell disruption improved protein digestibility without significant effect on the content of each amino acid, thus increased the protein digestibility-corrected amino acid score of each microalgae species. A larger magnitude of increase in protein digestibility-corrected amino acid score was observed in the mechanically ruptured AO relative to its undisrupted counterpart AO; however, the value was still much lower than the mechanically ruptured CV and CS. The increased content of lipids, starch, and fiber after mechanical cell rupture suggests that these biomolecules were not completely measurable in undisrupted biomass due to inadequate pre-treatment of the samples under the conditions employed in each specific analytical method. In lipid analysis, the samples were not pre-treated with acid hydrolysis, which might have resulted in an underestimation in the lipid content of the CV. The results also suggest that CV was more recalcitrant to solvent extraction than the CS and AO. Similar situations occurred in starch and fiber analyses, resulting in the underestimation of both nutrients in the CS and AO. The mechanical cell rupture broke up the cell wall at least in part so that nutrients were released from the inside or became more accessible by the analytical reagents or the digestive actions of the gastric and intestinal tract post ingestion. As such, the fat, starch, and fiber contents in the three disrupted microalgae biomasses may be considered close to or taken as the “true” values. This phenomenon was discussed in detail in a previous study [14].

In accordance with previous studies [13,19], microalgae CV, CS, and AO had high and different protein contents, and mechanical rupture did not cause a significant loss of protein or its constituent essential amino acids. The content of other nutrients was also largely different between species, supporting the observations in other studies [9,42]. In contrast to the effect on protein content, mechanical cell-rupturing yielded substantial effects on the measured contents of lipids, starch, and fiber of the three microalgae species. Nevertheless, the impact varied among the species, with interactions being noted between specific nutrients and microalgae species. These findings corroborate that the structure and composition of microalgae cell walls depend on species, which determine the efficiency of mechanical cell rupture. Accordingly, the values of lipids, starch, and fiber in the disrupted biomass of the three species are considered more reliable. In future studies, the analytical method should be carefully optimized for specific microalga species to ensure the true measurement of a given nutrient. 

Protein quality is vital when evaluating the nutritional value that a protein can provide. Several methods have been developed to assess the quality of a protein [43]. Amino acid score is one of the key indicators of protein quality and expressed as the ratio between the content of the first limiting amino acid of the test protein and the content of the same amino acid in a reference pattern of essential amino acids [44]. This reference pattern was based on the essential amino acid requirements of infant and different-age children as published by the FAO/WHO in 1985 [45] and updated several times since then [29,46]. The first limiting amino acid is the shortest supply or the single most limiting essential amino acid in digested protein relative to body requirements, and it first becomes deficient in the diet and thus limits the utilization of other amino acids in excess. A higher amino acid score means that the essential amino acids are more balanced and can be used more efficiently in the body for the synthesis of proteins and other amino acid-derived biomolecules. An amino acid score higher than 1 means that all essential amino acids of the test protein meet or exceed the requirements as defined in a given reference pattern, which depends on the targeted age group of the human population. In this regard, animal proteins deserve the highest rating because their amino acid compositions are comparable to the requirements of humans and animals [43]. The amino acid composition of casein used in the present study was similar to the product specification provided by the supplier and that reported in a previous study [47], demonstrating that the method of amino acid analysis was reliable. The first limiting amino acid of casein was methionine plus cysteine, which was the same as previously reported [31,36,47]. The amino acid score of casein was 1.26, which was higher than the values of 1.03–1.05 reported in some studies [31,36] while close to the value of 1.32 published by others [47]. Tryptophan was also reported as the first limiting amino acid of casein, with the amino acid score of 1.21 [34]. The differences in the first liming amino acid and amino acid score of casein between studies might be a result of variations in the amino acid analysis, source of casein, and the reference patterns used in different studies. Limited information is available regarding the first limiting amino acid and amino acid score of microalgae protein of different species for human nutrition. The present study demonstrated that histidine was the first limiting amino acid of the CV and AO, while isoleucine was the first limiting amino acid of the CS. The amino acid scores of the CV and CS were close to that of casein, but a lower amino acid score was seen in the AO. Mechanical cell rupture did not change the amino acid composition and score of the CV and CS. The slightly decreased amino acid score of the AO after mechanical cell rupture was caused by the decrease in the amount of the first limiting amino acid, histidine. As the content of other amino acids was not decreased, the reduction of histidine was considered to be a result of analytical variations.

Apparent protein digestibility reflects the amount of protein digested and absorbed and is used conventionally to assess the quality of a protein. However, the amount of endogenous metabolic protein excretion is not counted in the calculation of apparent protein digestibility. Advanced from apparent protein digestibility, true protein digestibility is corrected for the endogenous protein loss and measures the true amount of dietary protein that is digested and absorbed relative to the total amount ingested. The rat assay employed in the current study for measuring true protein digestibility was based on the Association of Official Agricultural Chemists Official Method 991.29 [30], which has been recommended by the FAO/WHO since 1989 [29,44,46]. In this method, male weanling rats with an initial weight of 50–70 g are used as the test animals, casein is used as a reference standard, and a protein-free diet is used to correct for endogenous protein loss. The rat assay has been used widely to assess the true protein digestibility of protein products, with different initial rat weights, such as 70 g [31,36,48], 150 g [49], and 250 g [50], being reported. In the present study, rats weighed approximately 156 g when they were started on the experimental diets. The true protein digestibility of the casein sample fell within the range of published studies using the same rat model and assay protocol [33,34,35,36,37,38,39], demonstrating the validity of the current study protocol and analytical methods. A 36–65% lower true protein digestibility of three microalgae species compared to casein suggests that the structures and constituents of microalgae interfered with the accessibility and action of digestive enzymes and fluids, which is in line with other studies [51,52]. Mechanical cell wall disruption dramatically improved the protein digestibility of each microalgae species. A similar funding has been reported in a previous study where ultra-sonication improved microalgae protein digestibility [22]. In addition, we observed that the “true” amounts of carbohydrate and fiber appeared to affect negatively the protein digestibility. The significant correlation between in vitro and in vivo protein digestibility suggests that the in vitro pepsin–pancreatin method, although originally optimized for salmon aquafeed research [23], may be equally suitable for estimating the protein digestibility of microalgae biomass for human and terrestrial animal food/feed applications and for screening product candidates during protein product development. 

It should be realized that the true protein digestibility determined by the rat assay and fecal nitrogen analysis is affected by several factors, particularly endogenous nitrogen loss and protein source. Several different techniques have been developed to measure endogenous protein loss, such as the use of a traditional protein-free diet, a highly digestible or enzymatically hydrolyzed protein-based diet, and the regression method [53]. There are criticisms with the protein-free diet method about the abnormal physiological state induced by severe amino acid deficiency; however, this method is still the most widely used method for measuring endogenous protein loss [53]. In addition, limitations with the fecal protein/nitrogen analysis method for measuring true protein digestibility are well recognized, and the ileal nitrogen analysis method is the preferred alternative [54] and promoted in recent years by the FAO/WHO [29,46]. However, due to technical difficulties, the fecal nitrogen analysis method is still used globally [36,55].

Among the several methods available for the final protein quality assessment, the protein digestibility-corrected amino acid score has been used the most widely [50]. The protein digestibility-corrected amino acid score is lower than 1 in most of the protein products, especially the plant-based proteins [31,36], including microalgae [56]. A product with the protein digestibility-corrected amino acid score exceeding 1, such as the casein tested in the present study, is not considered to contribute additional benefits in humans and is thus truncated to 1. The FAO/WHO Expert Consultation agreed that the protein digestibility-corrected amino acid score determined using the rat assay was a suitable approach for the routine evaluation of overall protein quality for humans [44]. The protein digestibility-corrected amino acid score was similar between the CV and CS but much lower in the AO, which is a result of lower values of both true protein digestibility and amino acid score. Mechanical disruption increased substantially the protein digestibility-corrected amino acid score of three microalgae species by improving protein digestibility. Compared to casein, three raw and mechanically ruptured microalgal species had lower protein digestibility-corrected amino acid scores and subsequently rats on diets containing one of these microalgae biomass as the sole source of protein had lower (or even negative) weight gain, while rats on the casein diet had a significant weight gain. To our knowledge, this is the first report on the comparison of protein digestibility and protein digestibility-corrected amino acid score between different microalgae species and the improvement by mechanical cell wall disruption using the rat assay and fecal nitrogen analysis. In 2020, a different assay protocol was used in rats to measure the protein and amino acid digestibility of a different microalgae species, cyanobacterium *Spirulina* [56]. In this study, ^15^ N recovery in digestive contents collected from caeca after 6 h of a single meal was used, with 86.0% of true protein digestibility and an 0.84 protein digestibility-corrected amino acid score being obtained, which were higher than the values of the three microalgae species assessed in the present study. In recent years, there is an increased interest in developing new sources of protein alternative to animal proteins, in particular, in the area of plant-based proteins, including pulses and algae [57,58]. The present study showed that the protein digestibility-corrected amino acid score of microalgae CV and CS was higher than that of lentils, beans, peas, and chickpeas [31,36], suggesting that the protein quality of these two microalgae species is better than pulses. In the past years, microalgae have been increasingly consumed mainly as whole cells for functional benefits beyond the traditional considerations of nutrition [59]. Therefore, the disruption of the cell wall using different technologies and selecting species with high inherent digestibility (e.g., low recalcitrance) has become crucial for developing microalgae proteins if consumed as whole cells. Generally, when protein is consumed in a pure form such as protein extract, the quality is enhanced dramatically as a result of increased digestibility [60]. However, information on the protein quality of microalgae protein concentrates or isolates are presently unavailable. Microalgae exploitation as a source of protein for the food industry still presents some drawbacks, mainly due to the lack of scalable and cost-effective production, harvesting, and downstream processing technologies [58]. It is believed that with advances in microalgae research and technology development, microalgae protein will be increasingly produced and used to meet the growing market demand for high-quality protein products. 

## 5. Conclusions

The results of the present study demonstrated that microalgae CV, CS, and AO were rich in protein. The CV and CS had high contents of essential amino acids and high amino acid scores, which were close to the amino acid score of casein, with histidine and isoleucine being the first limiting amino acids, respectively. However, the CV and CS had lower protein digestibility than casein and thus lower values of protein digestibility-corrected amino acid score. Mechanical cell rupture improved protein digestibility while having no significant impact on the protein and amino acid contents, resulting in a substantial improvement of protein quality. The AO had a low content of the essential and first limiting amino acid histidine, and therefore, the low values of amino acid score and protein digestibility-corrected amino acid score, even after mechanical cell rupture. The cell wall of microalgae was difficult to disrupt by in vitro and in vivo digestive systems and interfered with the protein digestibility and quality. Although mechanical cell rupture improved protein digestibility, the effectiveness still needs to be evaluated, and other methods such as the enzymatic method alone and in combination with mechanical disruption should be investigated. The application of cell-rupture technologies increases not only the digestibility of microalgae proteins when consumed as whole cells but also protein extraction efficiency when a more pure form of protein product is preferred. As a result of matrix effect or the interference of other components of microalgae such as carbohydrate and fiber, it is also important to assess the quality of microalgae proteins in a purified form such as protein concentrate or isolate and compare with the commercial protein products that are considered to have high quality, such as casein and soy protein isolates. As the in vitro method is fast, less expensive, and needless of animals, this method is valuable in screening protein product candidates at the early stage of product development. When lead candidates are finalized, an in vivo assessment is still required to generate the true protein digestibility-corrected amino acid score for product registration and regulatory approval.

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
