# Peer review of "A Rat Study to Evaluate the Protein Quality of Three Green Microalgal Species and the Impact of Mechanical Cell Wall Disruption"

_foods, 2020, doi:10.3390/foods9111531_

Round 1

Reviewer 1 Report

The works describes the protein quality of microalgae on the basis of amino acid composition and protein digestibility in rats. The authors compared 3 microalgae species and the effect of mechanical cell wall disruption.

 The chemical score of one of the three species (AO) revealed an insufficient content of histidine. The true protein digestibility was low: 38 to 75 % and was improved to 67 to 77 % after cell wall disruption. As a result, the PDCAAS was below 1 for the 3 species, even after mechanical disruption.

The paper is clear and well written. It brings interesting data on microalgae species since there are little in vivo studies. Most of the methods are convenient.

However, the rat assay has been done using the protocol for PER, i.e. with young rats and a limited protein diet (10 % instead of 20 % for this age). For digestibility studies, the rats are adult (weight of about 200-250 g), and the diet is adequate in protein (10 to 14 %). It’s amazing that the study has been conducted in deficient protein conditions. Can the authors justify this choice? In any case, they must highlight in the document that they used a different model than usually used. Even if they found a coherent value for casein, as it is a highly digestible protein, protocol errors marginally impact the result. In the present protocol, TPD was assessed with a protein free group. As rats are young and largely below their amino acid requirement, this must impact endogenous protein in the intestine.

Title: I would remove “apparently similar”

Abstract:

Specify the rat weight.

The conclusion of the abstract must be modulated. Indeed, the protein quality of microalgae is modest and thus the wording “promising source “sounds somewhat overrated. It would be better to qualify as “acceptable source” because the PDCAAS is around 0.8 (after pulverization).

Material and methods:

What was the acceptability of the products by the rats? What are the results of food intake? Animal growth is also a basic parameter to provide.

Can you describe how the rat assay was organized? 70 animals were allocated in 7 groups, what can hardly be realized in one experimental batch.

Par 2.1: What amount of algae biomass was produced?

For how long time the pulverization procedure of cells was applied? Was the temperature controlled? How the degree of disruption controlled?

Par 2.3: Nitrogen determination: please indicate the standard used. Idem for AA determination, which standards (internal and external).

Formula of AA Score: please indicate how was estimated the protein quantity (AA sum, N x 6.25?)

Table 1: there is a problem for the composition of CV/CVMR as the sum is > 100%. Starch seems very high, is it not erroneous?

Table 2: please indicate the sum of IAA and total AA

Is non-protein N can be determined, or at least estimated from the difference between the free AA sum and Nx6.15? Could you also please give the AA composition for 1 g of biomass, at least in supplemental file?

PDCAAS: please indicate the first limiting AA, and where appropriate the secondary limiting AA.

Discussion:

A paragraph on the limitation of the study must be added. Especially, discuss the impact of the specific model (i.e. young and prebubertal rats fed a low protein diet, below the requirements) on the results (cf my first comment).

 When the PDCASS is 1 (or more), there no limiting AA.  In consequence:

L378-379: Reword  as “…means that there are no limiting AA regarding the requirements.”

L387: Idem, Sulfur amino acids are not limiting in Humans. However, specify that there are in rats.

  390: tryptophan is not limiting AA. Please remove.

L 402-404: the sentence is not clear, especially the last part. Please clarify.

L 435-436. This could be modulated. Even in a different cell specie, there is a recent study on spirulina protein quality in rats (Tessier et al, Eur J Nutr 2020).

As asked above, please highlight and discuss the limiting AA(s) of the cell products.

Reviewer 2 Report

The manuscript by Wang et al. describes the impact of protein from different algae spices on protein digestibility. The topic is timely and relevant in the field of human nutrition, especially with the continuous search for alternative and less expensive source of proteins. In general, results support the conclusions that were presented. However, I have a few comments and suggestions that I hope the authors can address.

  • Title:

The title is not representative of the article content notably the in-vivo study.

  • Abstracts:

The abstract is clear, but too long. I suggest to reduce the length of the abstract by removing some information of the mat and meth section.

  • Material and methods

Line 122: proximate analysis of microalgal biomass and casein samples.

Line 158: The animal model is correct, however, the study period (8 days) is very short. Generally, the first week is dedicated to the adaptation of animals to experimental food. Therefore, the quantity of experimental food consumed could vary in a large extent during the first week. I suggest to add the food intake in the table 4.

Line 163: It is important to clarify the role of protein free diet in the model, and to discuss results further on the endogenous fecal nitrogen which could be different according the considered microalgal mass.

line179: Protein digestibility-corrected amino acid score (PDCAAS) is a method evaluating the quality of a protein based on the amino acid requirements for humans, and could not be calculated in animal models because the amino acid requirements of rats are different from those of humans.

Line 190: what is the method for nitrogen determination?

Line 234: please add a case for the total essential AA in table 2.

Line 235: Correlation is a statistic that characterizes the existence or absence of a relationship between two samples of values taken from the same group of subjects. Here, correlation is not relevant comparing the AA content of the different algae and casein as the groups are different from each other.

Line 269: all the AAS presented in table 3 (except for his in AO) are higher than 1 which means that all essential amino acids are adequate with human requirements. In this case met+cys could not be considered as a limiting amino acid in casein because it covers all body needs in sulfur amino acids. The same comment for other microalgae (CV and CS).

Line 284: AAS in table 4 is not required.

Line 327: The ability of dietary amino acids to be used by the body for maintenance and growth is an important parameter for protein quality determination. It would be interesting to evaluate the protein utilization using the urinary nitrogen.

Line 480: I suggest to add the energy intake in the supplementary materials (table 1).

How was the Nutrient density calculated notably lipid and starch?: The fat content reported in table 1 is similar in all diet (around 10%). However, the fat content of CV and CVMR is different. The same comment for the starch in AO and AOMR.

Reviewer 3 Report

Review of Manuscript Number: ID: Foods-928514 peer review – v1

Title:Protein quality is different among three apparently similar green microalgal species and improved  differentially by mechanical cell wall disruption

 Journal: Foods

The authors of the manuscript raise an important topic of research, which is an evaluate the protein quality three different microalgal species selected for high protein content and the impact of mechanical cell wall disruption on protein quality parameters.

The results suggest that the microalgae species promising sources of protein for humans and animals and quality can be markedly improved by  mechanical cell wall disruption. Additionally, in vitro protein digestibility measured using the  pepsin-pancreatin method may be used to screen protein product candidates, save animals, reduce  cost, and accelerate product development.

In conclusion, the manuscript is interesting. I suggest minor editorial changes:

line 98 - conditions for centrifuge – speed, time, temperature should be given

line 112 – 25-75 mL/min should be mL min-1

line 119 - what was the grinding time, to what particle size the test material was ground

line 154 – mg/g should be mg g-1

lime 143-144 - what kind of column, phases, were used in the analysis, what conditions

line 185 – mg/mL should be mg mL-1

line 209, 237, 269 - in tab. 1, 2, and 3 should include the standard deviation values

line 506 - a page range should be given

line 528 - page numbers should be completed

line 559 - page numbers should be verified 170-170

line 589, 591, 616, 626, 632 - a page range should be given

line 617 - is it a magazine or a book? give volume, pages, year

Round 2

Reviewer 2 Report

The authors responded to all comments, and significantly improved the manuscript.

Author Response

No new comments from the reviewers.